# Motivations for an Analytical RDF Database System

Christophe Callé[1,2], Philippe Calvez[2], Olivier Curé[1]

[1] LIGM Univ Paris Est Marne la Vallée, CNRS, F-77454.
{firstname.lastname}@univ-eiffel.fr
[2] ENGIE LAB CRIGEN philippe.calvez1@engie.com
{philippe.calvez1}@engie.com

**Abstract.** In this paper, we claim that by corresponding to an OLTP database management system, most RDF stores are addressing the wrong users. They should in fact consider the OLAP market. After motivating this position, we present a preliminary work toward developing a scalable analytical RDF database system on top of Apache Spark.

## 1 Introduction

Most production-ready RDF stores, *e.g.*, AllegroGraph[3], GraphDB[4], MarkLogic[5], AWS Neptune[6], Oracle[7], RDFox[8], Stardog[9], Virtuoso[10], are focusing on OLTP (online transaction processing) properties, *e.g.*, support for ACID transactions. But their workloads do not correspond to the processing of high rates of transactions. In fact, according to the most frequent use cases highlighted by these database management systems (DBMS), *e.g.*, Advanced search and discovery, Analytics/BI, Fraud detection, Recommendations, they should evolve in an analytical context. This OLAP (online analytical processing) orientation has been confirmed in [5] where it is considered that graph processing at large, *i.e.*, both the Labelled Property Graph and RDF data models, is mainly concerned with analytics since users are spending most of their time on testing, debugging, maintenance, ETL (Extract, Transform and Load) and cleaning.

One aspect that differentiates these two kinds of DBMS is the rate at which transactions are submitted. In OLTP, this rate is in order of hundreds to millions of transactions per second while in OLAP large sets of transactions are handled once per day, week or month. This typically corresponds to what we witness in the Linked Open Data context where data set updates can be released every month, *e.g.*, DBPedia. As a last argument toward this OLAP orientation, it is hard to believe that a OLTP database system does not support declarative update operations for close to 5 years. That is the time elapsed between the SPARLQ 1.0 and 1.1 which introduced INSERT and DELETE operations.

---

[3] https://allegrograph.com/products/allegrograph/

[4] https://graphdb.ontotext.com/

[5] https://www.marklogic.com/

[6] https://aws.amazon.com/fr/neptune/

[7] https://www.oracle.com/

[8] https://www.oxfordsemantic.tech/

[9] https://www.stardog.com/

[10] https://virtuoso.openlinksw.com/

Recently several production-ready systems started to introduce analytical operations in their offer, *e.g*., AnzoGraph DB[11] and Stardog as pure RDF store players and SANSA[3] as a library. We consider that most of the aforementioned systems will rapidly propose analytical features. Obviously, this will help the emergence of innovative functionalities and query processing performance improvements.

In the following, we present an analytical direction and motivate the adoption of a popular Big data framework for developing an open source, vertically scalable analytical RDF store. In a preliminary experiment, we outline the storage benefits that Spark offers for conducting analytical operations.

## 2   RDF stores and data analytics

Analytic operations in an RDF store, or any graph data model, can take the following forms: cube-inspired or graph algorithm-based. The former correspond to what one can find in relational OLAP (ROLAP) systems with their drill-down, roll-up, dice, slice and pivot SLQ extensions. The integration of such analytical operations has been considered in [2] where it is assumed that the RDF DBMS manages and stores transactions. We do not consider that it makes sense and that these data should remain in a real OTLP DBMS. The latter analytical kind has been confirmed to be the most relevant for graph stores in [5]'s survey. In fact, the identified top graph computations are finding connected components, neighborhood queries, finding shortest paths, ranking and centrality scores, reachability queries, triangle count and enumeration.

Out of the three analytical RDF systems that we have previously mentioned, AnzoGraph DB can be considered as the first and most complete analytical system. This commercial database system partially proposes both kinds of analytics (cube and graph algorithms). Stardog is a well-established commercial RDF stores that first started to propose graph analytics in its latest version (*i.e*., 7.5 as of march 2021). Finally, SANSA consists of a set of open-source libraries to perform reasoning, querying and analytic operations. Considering analytics, it relies on either Apache Spark[12] or Apache Flink[13] distributed computing frameworks. Some limitations of SANSA is that it requires a programmer to design an application on top of these libraries. So the system may be difficult to use out of the box by a data analyst. Moreover, the integration of graph algorithms within SPARQL queries is non declarative and thus requires some programming.

Similarly to SANSA, we are aiming to design our analytical RDF store on top of Apache Spark. This full-featured open-source parallel computing engine is an interesting platform to implement our system on top of. It proposes mature, extensible query processing, *i.e*., SQL and DataFrame (DF) Domain Specific Language (DSL), and optimization, *i.e*., catalyst, components on which SPARQL processing can benefit. Moreover, Spark is evolving toward a data lake and so-called lakehouse[1] approaches by leveraging from its large machine learning, stream and graph algorithm processing capacities. It is important to note that apart from SANSA, the Semantic Web community has already used Spark to implement RDF DBMS, among them [6], [4].

---

[11] https://www.cambridgesemantics.com/anzograph/

[12] https://spark.apache.org/

[13] https://flink.apache.org/

The data storage aspect plays an important role in the design of an OLAP database system, independently of the underlying data model. Apache Spark is offering many advantages with the native support of the most popular hybrid columnar storage systems, namely Apache Parquet[14] and Apache ORC[15] (Optimized Row Columnar). These two data formats provide a high compression rate together with efficient compression/decompression algorithms which are optimized in Spark, particularly for Parquet which is the default format for Spark's main data abstraction, *i.e.*, DF. This is especially important in a Spark context where the data is supposed to remain in main memory as much as possible. Thus the more you can fit in the main-memory, the less spill to secondary storage and thus the less Input/Output. In addition, the application programming style encouraged in Spark facilitates the creation and manipulation of DF from over DF. The created can be considered as materialized views which are frequently encountered in ROLAP systems. The computation cost of these views is largely amortized in an ecosystem where the rate of transaction is in terms of days to months.

## 3 Preliminary experimentation

We are conducting all experimentation with a synthetic Lehigh University Benchmark (LUBM)[16] instance (henceforth denoted LUBM1000) configured with 1.000 universities, *i.e.*, 1.383 million triples. The computing setting is a single Dell PowerEdge R740XD equipped with an Intel Xeon Gold 6230. By default, we allocate 16 cores to each jobs and 64GB of RAM for the Spark's driver. It runs Apache Spark 3.1.1 with scala 2.12.9 and Java 11.

The original file size for LUBM1000 is 23GB in the N-Triples syntax. In Table 1, we compare the two binary data formats previously mentioned (Apache Parquet and Apache ORC) with a CSV serialization. That table clearly highlights some valuable properties. Parquet and ORC provide a high compression rate compared to the original data size (and CSV format), respectively compressing to 6.95% and 7.8%. Moreover, this high compression does not impact the time performance of writing and reading the data sets. In fact, considering the writing operation Parquet is 1.15 times faster than CSV which is itself 1.19 times faster than ORC. Parquet and ORC are both faster to load than CSV, taking around 41% of the time required by the latter.

**Table 1.** Storing LUBM1000 (16 partitions)

| Serialization | Size (GB) | Creation duration (ms) | Loading duration (ms) |
|---|---|---|---|
| CSV | 23 | 47.287 | 13.170 |
| Apache Parquet | 1,8 | 40.826 | 5.407 |
| Apache ORC | 1,6 | 56.275 | 5.472 |

---

[14] https://parquet.apache.org/

[15] https://orc.apache.org/

[16] http://swat.cse.lehigh.edu/projects/lubm/

The high compression rate, together with the fast compression/decompression times, are important properties in Spark where the goal is to keep the data set in main memory as much as possible. ORC is notoriously recognized to be more compact than Parquet. Nevertheless, the creation and loading times emphasize the fact Parquet is the preferred format in Spark, with more optimization being implemented than for ORC. Moreover, the importance of the number of partitions, of key importance in Spark processing, also has an impact on Parquet's storage. Hence, with 4 and 8 partitions, the sizes are respectively of 1.5GB and 1.6GB (compared to 1.8GB for 16 partitions).

We now consider a naive, *i.e.*, with no specific query optimization, processing of a simple aggregation queries. In general, OLAP database systems compute materialized views for frequent queries. The DF DSL enables to create and persist intermediate query results in a DF which can itself be queried. In the case of our simple query (counting the number of courses taught by each professor type), we first materialize this DF (view) in about 9.5 seconds (for over 2.1 million records). Then we compute several aggregations from this query in about 1 second.

## 4   Conclusion

In this short paper, we have emphasized that RDF stores should migrate from the OLTP to the OLAP market. Some innovative systems have recently adopted this trend and we believe that most database management systems aiming to survive in the RDF and graph ecosystems will go in this direction. Looking at the craze for data analytics in the relational data model, where research has been around for a while, we understand that there is a lot of research to be done towards an analytical RDF store. We presented an approach based on the Apache Spark engine and provided some encouraging preliminary results on the data storage aspect. This will lead the way to some future research and implementation work.

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
