# OpenReview forum: "Motivations for an Analytical RDF Database System"
_eswc-conferences.org/ESWC/2021/Conference/Poster_and_Demo_Track — Submitted to ESWC2021 P&D_

### Official Review · AnonReviewer2 · 2021-04-14
**Good motivation, but it seems there is already a good body of work on the topic**

**Rating:** 3
**Confidence:** 4

**Review:**

The paper makes a claim that RDF stores shouldn't follow the OLTP database approach, but rather consider the OLAP market. The paper points the main differences in types and frequency of transactions in both RDF stores and OLTP databases, and how RDF stores would benefit from an OLAP approach.

It goes on to present their early work on building an RDF store on top of Apache Spark, showing preliminary results on the data compression aspect (which allows to fit more data in the main memory).

I think the arguments are convincing for a vision paper. What is unclear, however, is how much of a "vision" it really is. The authors point out that OLAP RDF stores have in fact been the subject of study in previous work, with systems like AnzoGraph and Stardog supporting analytical operations, as well as the SANSA library.  Moreover, the work in [4] and  [6] already exploit Apache Spark.

I think the idea is interesting, but I think the novelty/vision needs to be improved to make it clear how they differ from SOTA.

**Anonymity:**

Yes, I would like my review to remain anonymous.

---

### Official Review · ~Marilena_Daquino1 · 2021-04-14
**interesting insights on a relevant topic, not enough evidence to accept the conclusions**

**Rating:** 6
**Confidence:** 2

**Review:**

quality
- the claims and the motivations to use Apache Spark are generally sound. However, the preliminary demonstration is not extremely insightful since it focuses only one (relevant) of the features to be addressed.

clarity
- some technical details are unnecessary for the sake of the narrative
- some aspects could be elaborated more, e.g. “we are aiming to design our analytical RDF store on top of Apache Spark”. What is the goal of your work? If the poster is accepted, I’d frame better your work, so that it is easier to accept your conclusions as valid in a certain context, which currently cannot be generalised since (a) not all projects may have the same requirements and (b) the preliminary demonstration is not enough.

originality
- moderately original, falls under “technical report” rather than research paper/poster

significance
- the topic is potentially relevant to many communities that leverage graph data and ML.

pros
- highlights an interesting topic
- provides an overview of existing RDF DBMS
- provides a few insights on aspects that can be measured to evaluate RDF DBMS

cons
- the demonstration is too partial to generalize the conclusion (Spark is better)
- some claims are unjustified, e.g. “We do not consider that it makes sense and that these data should remain in a real OTLP DBMS”, “AnzoGraph DB can be considered as the first and most complete analytical system.”


**Anonymity:**

No, I would like my review to be deanonymized.

---

### Official Review · AnonReviewer4 · 2021-04-15
**Lack of a tack away message**

**Rating:** 4
**Confidence:** 5

**Review:**

[Submitted by a subreviewer]

The paper proposed an interesting “position” and emphasize that the RDF stores should migrate from the traditional OLTP model more towards the OLAP market. Even though this approach has been already addressed by some of the techniques mentioned in the paper, the position brings up here is an interesting topic to be introduced within the community. Later, it also presents some of the very short preliminary results on such a proposal; scalable analytical RDF database system builds on top of the distributed computing framework – Apache Spark.

Strong Points:
- (+) the topic is interesting
- (+) the motivation is well justified
- (+) the paper is well written and organized
Negative Points:
- (-) it doesn’t provide proper comprehensive analysis and evaluation
- (-) it lacks comparison with other well-known distributed analytics framework (mentioned in the paper)
- (-) the scalability is not well supported, or at least shown on the paper (mentioned that the experiments were run on a single machine vary with its core)
- (-) the source code and instructions for reproducing the results are not available online

Overall, the paper is well-written and structured. The motivation for having an analytical distributed RDF database system ported on the OLAP marked is well placed.
However, in its present form, the paper suffers from a number of major issues, which are outlined below.
- I give a good overview of the limitations and mention some of the existing approaches, however the details which support such claims are missing. There isn’t any architecture overview presented that tells us how such analytics can be performed using Apache Spark? Besides some info on the storage level (compression mechanism) there isn’t anything explain in the paper. I do understand that due to the page limit the paper had to go through a “brute force” shortened, but such information is evident when proposing such an approach.
- I see that there are some preliminary results already reported on the paper, but they lack comparison with other approaches (i.e. SANSA Sparklify/Ontop, or SPARQLGX-SDE) but rather it only reports on the run time for read/write for different io serialization formats. None of the LUBM query suites has been evaluated?

As a result, with the current version of a paper I do not see a clear “take away” of what it does differ from other existing solutions and the benefits of using this approach, mostly due to the lack of architectural details, implementation details, source-code/documentation, evaluation/benchmarking, etc)

**Anonymity:**

Yes, I would like my review to remain anonymous.

---

### Official Review · AnonReviewer1 · 2021-04-15
**Scalable analytical RDF database system using Apache Spark**

**Rating:** 5
**Confidence:** 3

**Review:**

The paper is focused on a preliminary study for developing a scalable analytical RDF database system using Apache Spark.


- I am very unclear about what is added advantage of this paper as compared to SANSA stack.
- I am wondering about the novelty in this paper.
- The paper only discusses some preliminary experiments in the last two pages the rest seem like a description. A figure of the architecture would be nice.
- An example of the problem would be really helpful.

**Anonymity:**

Yes, I would like my review to remain anonymous.

---

### Decision · Program_Chairs · 2021-04-19

Reject